# Influence of Cultivation Conditions on the Sioxanthin Content and Antioxidative Protection Effect of a Crude Extract from the Vegetative Mycelium of *Salinispora tropica*

**DOI:** 10.3390/md19090509

**Published:** 2021-09-08

**Authors:** Zuzana Jezkova, Vera Schulzova, Ivana Krizova, Marcel Karabin, Tomas Branyik

**Affiliations:** 1Department of Biotechnology, University of Chemistry and Technology Prague, Technicka 5, 166 28 Prague, Czech Republic; jezkovaz@vscht.cz (Z.J.); krizovaa@vscht.cz (I.K.); karabinm@vscht.cz (M.K.); 2Department of Food Analysis and Nutrition, University of Chemistry and Technology Prague, Technicka 5, 166 28 Prague, Czech Republic; schulzov@vscht.cz; 3Research Institute of Brewing and Malting, Lipova 15, 120 00 Prague, Czech Republic

**Keywords:** *Salinispora tropica*, total cellular carotenoids, sioxanthin, antioxidant activity

## Abstract

Due to their bioavailability, glycosylated carotenoids may have interesting biological effects. Sioxanthin, as a representative of this type of carotenoid, has been identified in marine actinomycetes of the genus *Salinispora*. This study evaluates, for the first time, the effect of cultivation temperature (T) and light intensity (LI) on the total cellular carotenoid content (TC), antioxidant activity (AA) and sioxanthin content (SX) of a crude extract (CE) from *Salinispora tropica* biomass in its vegetative state. Treatment-related differences in TC and SX values were statistically significantly and positively affected by T and LI, while AA was most significantly affected by T. In the *S. tropica* CE, TC correlated well (R^2^ = 0.823) with SX and somewhat less with AA (R^2^ = 0.777). A correlation between AA and SX was found to be less significant (R^2^ = 0.731). The most significant protective effect against oxidative stress was identified in the CE extracted from *S. tropica* biomass grown at the highest T and LI (CE-C), as was demonstrated using LNCaP and KYSE-30 human cell lines. The CE showed no cytotoxicity against LNCaP and KYSE-30 cell lines.

## 1. Introduction

Carotenoids are pigments of natural origin created by the terpene biosynthetic pathway. They occur in various microorganisms, plants, and animals. Attention is paid to them mainly on the basis of their potential beneficial effects on human health [1]. They have strong antioxidant activity and are able to scavenge free radicals from the environment, thus being effective in preventing degenerative diseases. Their function depends on their molecular structure, in particular on the number of conjugated double bonds, the types of functional groups, and their molecular weight [2,3].

Marine ecosystems are associated with a great diversity of species, which may represent an attractive source of novel biologically active substances. Marine carotenoids have shown antioxidant properties in reducing oxidative stress markers and their mechanism to quench O_2_ and scavenge free radicals has been proposed [4].

Marine actinomycetes of the genus *Salinispora* are capable of producing orange carotenoids in the vegetative phase of growth [5]. In the case of *Salinispora tropica* CNB-440^T^, one of the orange carotenoids was identified as sioxanthin (SX) and its structure was determined [6]. This substance is one of the newly described marine carotenoids, belonging to rare glycosylated carotenoids, which have not found therapeutic application so far. However, the potential effect of SX has not yet been fully investigated. It is a C40 carotenoid that has a glycosylated bond at one end of the chain and an aryl group at the other. Its structure determines the unusual amphiphilic character of the molecule, which can also affect its biological activity, in particular through increased bioavailability [7,8]. Considering the presence of conjugated bonds in the SX molecule and the function of other important carotenoids (beta-carotene, lycopene, or lutein), similar mechanism of action and strong antioxidant activity may also be expected [9].

Because *S. tropica* carotenoids are intracellular metabolites, their formation is associated with biomass growth. Cultivation conditions (temperature, oxygen supply) and medium composition play an important role in biomass productivity of *S. tropica* [10]. The composition of the culture medium also influences the formation of carotenoids [11]. In general, light induces carotenogenesis in microorganisms, leading to their increased production [12]. In the microalga *Haematococcus pluvialis*, increased light intensity was shown to regulate the level of phytoene synthase and carotenoid hydrolase [13]. In the marine bacterium *Formosa sp.*, an effect of light conditions on biomass production was not observed, but a positive effect on carotenoid production was evident [14].

Therefore, this study aims to evaluate the effect of cultivation conditions such as temperature (T) and light intensity (LI) on the total cellular carotenoid (TC) content of *S. tropica* biomass obtained during the vegetative stage of growth. Moreover, antioxidant activity (AA) and sioxanthin (SX) content in the crude extract (CE) of *S. tropica* biomass was determined. The effects of individual parameters on carotenoid composition in the *S. tropica* crude extract (CE), as well as CE antioxidant activity, were statistically evaluated. The protective effect of *S. tropica* CE against oxidative stress induced by ROS and the corresponding cytotoxicity was tested using different human cell lines. This study is expected to provide baseline information for further exploitation of *S. tropica* biomass as a source of protective antioxidants.

## 2. Results

### 2.1. Identification of Sioxanthin

LC/MS was used for the identification of SX, a compound with the systemic name (2′S)-1′-(β-D-glucopyranosyloxy)-3′,4′-didehydro-1′,2′-dihydro-φ,ψ-caroten-2′-ol [6]. SX was detected by DAD as a peak with a retention time of 3.0 min (Figure 1). These spectra and the peak areas of SX were used to evaluate the effect of T and LI on the SX content of CE from *S. tropica*. Identification of SX was carried out both by the UV–VIS spectrum of this peak with three absorption maxima at 450, 476, and 508 nm (Figure 1) and MS/MS spectra (Figure 2). The chemical formula of SX is C_46_H_62_O_7_. The identification of the target compound was based on its cation radical molecular ion [M]^+^ at *m/z* 726.4484 (Δppm = −1.58) and a fragment typical of carotenoids with an OH group in its molecule [M+H-H_2_O]^+^ at *m/z* 709.4447 (Δppm = −2.96).

### 2.2. Effect of Cultivation Conditions on the Crude Extract

Shake flask cultures of *S. tropica* were carried out at different temperatures (T, 20, 24, and 28 °C) and light intensities (LI, 0, 9.3, and 18.6 µE/m^2^/s). The harvested biomass was extracted and the CE was characterized by TC content, AA and peak area of SX, identified according to [6].

Changes in cultivation conditions resulted in different TC contents (Table 1). In general, the TC in CE were positively affected by both T and LI, which is also reflected in the shape of the response surface (Figure 3A). The ANOVA analysis showed that the quadratic model for TC of *S. tropica* was highly significant. The correlation between experimental values and response surface was R^2^ = 0.999. Multiple regression analysis of the experimental data generated a quadratic equation that demonstrated the high significance (*p*-value ˂ 0.03) for each term (T, LI, T^2^, and LI^2^) in predicting the TC response. The statistically most significant effect was found for term T (*p*-value = 0.0074) and LI (*p*-value = 0.012). The final equation for TC of *S. tropica*, expressed in terms of actual factors, is
TC = −3540.1 + 309.23 × T − 5.73 × T^2^ + 24.51 × LI − 0.82 × LI^2^(1)

The effect of cultivation condition on AA of the CE from *S. tropica* had an increasing trend with T (Table 1). The ANOVA analysis showed that the linear model for AA was highly significant, indicating an effect of T (Figure 3B). The correlation between experimental values and response surface was R^2^ = 0.971. Multiple regression analysis of the experimental data generated a linear equation that demonstrated that T had the most significant (*p*-value = 0.045) effect on AA, while the other terms were statistically not significant (*p*-value ˃ 0.1). The final equation for AA of *S. tropica*, expressed in terms of actual factors, is
AA = −1880.5 + 221.62 × T − 64.34 × LI + 2.17 × T × LI(2)

As the SX standard was not available, evaluation of the effect of T and LI was carried out with the peak area of SX. The highest SX peak area was obtained in CE from *S. tropica* biomass grown at the highest T (28 °C) and LI (18.6 µE/m^2^/s). The ANOVA analysis showed that the linear model for SX content in *S. tropica* CE was highly significant (Figure 3C). The correlation between experimental values and response surface was R^2^ = 0.999. Multiple regression analysis of the experimental data generated a linear equation that demonstrated that both T (*p*-value = 0.002) and LI (*p*-value = 0.0072) had highly significant effects on SX. The interactive term (T × LI) was also found to be statistically significant (*p*-value = 0.0089). The final equation for SX of *S. tropica*, expressed in terms of actual factors, is
SX = −346.62 + 19.85 × T − 38.39 × LI + 1.935 × T × LI(3)

It is clear that, with more than 95% confidence (*p*-value = 0.044), TC correlated (R^2^ = 0.823) with SX in CE from *S. tropica* biomass grown under different conditions. Simultaneously, the AA of CE correlated significantly less with TC (*p*-value = 0.069, R^2^ = 0.777) and SX (*p*-value = 0.099, R^2^ = 0.731). This indicates that the AA is not determined only by carotenoids in CE.

### 2.3. Protective Effects of Crude Extracts against Oxidative Stress

CE obtained from *S. tropica* biomass grown under three different conditions (Table 1) showed no protective effect in the case of cell lines HEK-293, KYSE-180, and HeLa (data not shown). A moderate protective effect of CE at concentrations of hydrogen peroxide ˃2000 µM was observed in the case of HEP-G2 and MCF-7 cell lines (data not shown). The most significant effect against oxidative stress induced by ROS was observed for human prostate adenocarcinoma (LNCaP) and esophageal squamous carcinoma cell lines (KYSE-30). The protective effect of CE-A (20 °C, 0 µE/m^2^/s), CE-B (24 °C, 9.3 µE/m^2^/s), and CE-C (28 °C, 18.6 µE/m^2^/s) is apparent from the decreased relative fluorescence of the stress indicator (CM-H2DCFDA) in the presence of H_2_O_2_ for both cell lines tested (Figure 4 and Figure 5). The stress indicator was oxidized by ROS to its fluorescent form in the absence of antioxidant protection by CE. At H_2_O_2_ concentrations 1000–2000 µM, the highest level of protection for LNCaP cells was provided by CE-B (0.2 ng/µL) and CE-C (0.2 ng/µL) (Figure 4). At maximum H_2_O_2_ addition (2500 µM) the LNCaP cells were most significantly protected by CE-C (0.2 ng/µL). In the case of LNCaP cell lines, the protective effect of β-carotene was lower than that of all CE tested (Figure 4). In the case of KYSE-30 cells, the protective effect of CE-C (0.2 ng/µL) and β-carotene (0.1 ng/µL) was comparable at H_2_O_2_ concentrations of 500 and 1500 µM (Figure 5). However, at higher H_2_O_2_ concentrations (2000–2500 µM) protection against oxidative stress by CE-C (0.2 ng/µL) was the most significant (Figure 5). In the absence of hydrogen peroxide, the fluorescence of oxidative stress indicator increased in the presence on CE compared to 1% DMSO and β-carotene (Figure 4 and Figure 5).

### 2.4. Cytotoxicity of Crude Extract

Cell viability was monitored in response to β-carotene (BC), CE-A (20 °C, 0 µE/m^2^/s), CE-B (24 °C, 9.3 µE/m^2^/s), CE-C (28 °C, 18.6 µE/m^2^/s), and pure DMSO solvent using the redox chromogenic dye resazurin. Blue resazurin is enzymatically reduced to red resorufin within a living cell, altering the absorption maximum based on which it is detected. The experiments were performed on the LNCaP and KYSE-30 cancer cell lines. Data were processed as the percentage survival of cells affected by BC, CE, and DMSO versus cells without exposure to these substances. The results show that 100% viability of LNCaP cells was preserved in all cases tested (data not shown). KYSE-30 cells showed negative responses (92.8%, 83.7%, and 71.26% cell viability) to increasing DMSO concentrations (0.5%, 0.75%, and 1% DMSO), respectively. The addition of BC did not affect significantly the negative effect of DMSO. The lowest KYSE-30 cell viabilities of 67.6% were observed for β-carotene (0.2 ng/μL in 1% DMSO). However, this viability value was statistically not different from 1% DMSO. Statistically higher KYSE-30 cell viabilities were observed for CE-A (94.7%, 0.1 ng/μL in 0.5% DMSO), and CE-C (95.5%, 0.1 ng/μL in 0.5% DMSO) as compared to pure 0.5% DMSO, indicating to protective effect of CE. For all the other CE in DMSO the KYSE-30 cell viabilities were clearly affected only by DMSO (data not shown).

## 3. Discussion

Many compounds have been isolated from the *Salinospora* genus. Some of them, such as salinisporamide A, have cytotoxic activity against carcinoma cells [15,16,17], while the biological activity of others, such as carotenoids, have not been characterized. In general, carotenoids are known to have a broad range of biological activities and thus they play an important role in the food, feed, cosmetic, and pharmaceutical industries [18]. However, the full utilization of their biological activity is sometimes problematic due to their low water solubility. This focused attention on uncommon glycosylated carotenoids. It has been shown that crocetin, a glycosylated carotenoid, has higher cytotoxicity against cancer cell lines and has a different mechanism of action compared with crocin, a carboxylic carotenoid [19]. The glycosylated carotenoid myxoxanthophyll was also found in the cyanobacterium *Synechocystis* sp. [20], and exhibited immune-stimulating properties by activating human granulocytes [21].

Glycosylated carotenoid SX has also been identified in marine actinomycetes such as *Salinispora*. However, industrial applications of carotenoids depend on the possibility of their large-scale production, the feasibility of which requires knowledge of the influence of process parameters on carotenoid content. The contents of TC and SX in *S. tropica* biomass were found to be temperature- and photo-induced. With the combined effects of T and LI, the TC content of *S. tropica* biomass increased by 125% from 354 µg/g (CE-A) to 800 µg/g (CE-C). Simultaneously, the SX peak area increased ten-fold and this was accompanied by a 75% increase in AA of CE. To our knowledge, glycosylated carotenoids of microbial origin have not been commercially exploited because potential scale-up of production requires tools for increasing productivity.

One of the roles of carotenoids within cells is to scavenge free radicals produced upon illumination, and the photoinduction of carotenogenesis is known in several prokaryotes including actinomycetes. *Streptomyces coelicolor* A3 (2) produced carotenoids (including β-carotene) under blue light induction and a light-induced sigma factor was identified that directs the transcription of the carotenoid biosynthesis gene cluster [22]. Besides actinomycetes, the carotenoid extract yield and carotenoid concentration were also higher when the marine bacterium *Formosa* sp. was grown under light [14].

Among environmental stimulants for the production of carotenoids by microorganisms, T has a rather variable effect [12]. For *S. tropica*, the optimum T for growth was found to be 28–30 °C [10], which is the T at which most TC and SX were found in CE. Although T-induction of carotenogenesis in actinomycetes has not been studied, there are examples of temperature induced carotenoid production by different species of microalgae and fungi [23,24]. In this work, it seems that T positively affects not only the growth rate of *S. tropica*, but also the activity of enzymes involved in the biosynthesis of carotenoids.

Oxidative stress results from an imbalance in the production of reactive oxygen species (ROS) and the ability of the cell to scavenge them. ROS reacts with nucleic acids, proteins and lipids causing cell and tissue damage and can be measured using selective or general indicators. The non-fluorescent dye (CM-H2DCFDA) becomes fluorescent green when oxidized by H_2_O_2_ and its free radical products. The ability of CE to reduce oxidative stress was tested on different human cell lines in order to define the potential applicability of *S. tropica* biomass.

The results showed that CE-C (28 °C, 18.6 µE/m^2^/s), with the highest TC, AA, and SX, provided the most significant protective effect against high concentrations of H_2_O_2_ (2000–2500 µM) in the case of both LNCaP and KYSE-30 cell lines. A similar protective effect was found for two actinomycete extracts against CCl4-induced liver damage in rats. This protective effect occurred through the reduction of oxidative stress and improvement of antioxidant defense markers [25]. Activity of enzymes involved in the detoxification of ROS was observed in methanolic extracts from *Streptomyces* sp. strain DBT34 [26]. The effect of CE obtained in this work could be ascribed to the well-known AA of carotenoids [27]. However, carotenoids are not the only compounds in microorganisms with AA [28]. This is supported by the fact that TC of CE from *S. tropica* correlates better with SX than does AA with both TC and SX. This finding supports the hypothesis that there are other, as yet unidentified, compounds with antioxidant activity in CE from *S. tropica*, as well as known AA compounds found in *Salinispora* species, including desferrioxamine B and E [7]. Therefore, further characterization of CE from *S. tropica* should be carried out.

## 4. Materials and Methods

### 4.1. Microorganism

*Salinispora tropica* CBN-440^T^ (DSM 44818, ATCC BAA-916) is deposited at the culture collection of the Leibniz Institute DSMZ-German Collection of Microorganisms and Cell Cultures (Braunschweig, Germany). The strain was obtained in freeze-dried form and maintained in cryovials (T310-2A 2 mL, Simport, Beloeil, QC, Canada) prepared from fresh cultures and stored at −70 °C.

### 4.2. Cells Lines

HEK-293 (ATCC CRL-1573™), HeLa (ATCC CRM-CCL-2™), HEP-G2 (ATCC HB-8065™), LNCaP (ATCC CRL-1740™), MCF-7 (ATCC HTB-22™) cells lines were obtained from ATCC^®^ (Manassas, VA, USA). KYSE-180 (ACC 379) cell line was from DSMZ (Braunschweig, Germany), and KYSE-30 (94072011) cell line was obtained from Sigma-Aldrich (Schaffhausen, Switzerland).

### 4.3. Media

Nutrient liquid medium (NLM) contained in g/L: 10 glucose (Penta, Prague, Czech Republic), 4 yeast extract (Carl Roth, Karlsruhe, Germany), 2 peptone from meat (Carl Roth, Karlsruhe, Germany), and 30 artificial sea salt (Royal Nature, Nesher, Israel). Nutrient solid medium (NSM) contained, in addition, 15 g/L bacteriological agar (Oxoid, Hampshire, England, UK). Soy medium (SM) contained in g/L: 10 glucose (Penta, Prague, Czech Republic), 7.31 soy peptone (Sigma-Aldrich, Schaffhausen, Switzerland) and 30 artificial sea salt (Royal Nature, Nesher, Israel). All media were adjusted to pH 7.0 ± 0.2 with 1 M NaOH or H_2_SO_4_.

### 4.4. Shake Flask

The preparation of cryovials and their use to seed the inoculum cultures is described in [10]. The inoculum cultures in NLM were then used to inoculate sterile SM at 10% *v/v* of the total volume. Shake flask cultures were grown five days into stationary phase in 500 mL baffled Erlenmeyer flasks (bottom diameter 100 mm, height 175 mm, neck diameter 29 mm) containing 150 mL of SM. Flasks were shaken at 150 rpm on a rotary shaker. Cultivations took place under changing conditions at 20, 24, and 28 °C and light intensity (0, 9.3, 18.6 µE/m^2^/s) inside empty flasks measured with a light sensor (QSL-2101, Biospherical Instruments Inc., San Diego, CA, USA). Shake flask experiments were carried out in triplicate.

### 4.5. Extraction of Biomass

Biomass suspensions were centrifuged (6000 rpm for 20 min, Sorwall, Thermo Scientific, Berlin, Germany), washed with distilled water (half of the volume of suspension) and centrifuged again under the same conditions. The biomass pellet was frozen (−70 °C, 12 h), covered with aluminum foil and placed in a lyophilizer (Heto PowerDry LL3000, Bath, England, UK). Lyophilization was performed at a reduced pressure of 4.10^−4^ bar and a temperature of −55 °C to constant weight.

To 100 mg of lyophilized biomass, 2 g of glass beads (1.0 mm in diameter, Zirconia, BioSpecProducts, San Diego, CA, USA) and 1 mL of demineralized water were added. The mixture was vortexed for 10 min. Then, 2 mL of dichloromethane: methanol (1:1) were added and the mixture was vortexed (2 min) and centrifuged (10 min, 6000 rpm). The liquid phase was collected, 1 mL of a dichloromethane: methanol (1:1) was added, the mixture was vortexed (2 min) and centrifuged (10 min, 6000 rpm) again. The liquid phase was collected and added to the previous extract. The extracts were filled with methanol to a volume of 5 mL. The combined CE were centrifuged (10 min, 6000 rpm) to remove residual biomass.

### 4.6. Total Cellular Carotenoids

Total cellular carotenoids (TC) were measured on a UV–VIS spectrophotometer at a wavelength of 475 nm in CE from *S. tropica* biomass. The resulting value was expressed as the amount of TC in µg/g of lyophilized biomass according to the relation
TC = (1/m) × (A/ε) × (V_r_/W),(4)
where m is the mass of biomass (g), A is absorbance, ε is the average extinction coefficient of carotenoid pigments 250 dm^3^⸱g^−1^⸱cm^−1^ [1], Vr is the volume of solvent (dm^3^) and W is the width of the cuvette (cm).

### 4.7. Antioxidant Activity

The diphenyl picrylhydrazyl (DPPH) method was carried out according to [29] with some modifications. The extracts from *S. tropica* biomass were allowed to react with DPPH solution for 30 min in the dark. The external standard calibration curve was linear between 200 and 800 µM Trolox and the AA was expressed in µM Trolox/g dry biomass.

### 4.8. Identification of Sioxanthin (LC/MS)

Crude extracts cultured under different conditions were evaporated under nitrogen. The dry extract was resuspended in 1 mL of acetone/ethanol (4:6) with 0.2% BHT. The sample was filtered through a 0.22 μm PTFE filter, before analysis. Extracts were measured using a UHPLC/DAD/MSD Q-TOF system (Agilent 1290 Infinity LC System with DAD Detector and Ion Mobility Q-TOF Mass Detector, Santa Clara, CA, USA). The measurement was performed on the same column under the same conditions as the measurement of sioxanthin on HPLC/DAD. MS conditions for positive mode of electron spray ionization: mass range 100–1700 *m/z*, nozzle voltage 1000 V, auxiliary gas 350 °C, auxiliary gas flow 11 L/min, nebulizer 35 psig, capillary 4000 V, fragmentor 400 V, drying gas nitrogen 8 L/min, drying gas temperature 350 °C, and collision energy 20 V. The identification of sioxanthin was carried out based on previously published data [6].

### 4.9. Determination of Sioxanthin (HPLC/DAD)

Crude extracts (5 mL) cultured under different conditions were evaporated under nitrogen. The dry extract was resuspended in 1 mL of acetone/ethanol (4:6) with 0.2% BHT. The sample was filtered through a 0.22 μm PTFE filter, before analysis. Extracts were analyzed using a high-performance liquid chromatography coupled with a diode array detector (HPLC DAD, Agilent Technologies 1200 series, Santa Clara, CA, USA) on a Poroshell 120 EC-C18 column (100 × 2.1 mm; 2.7 μm). The mobile phase was composed of 100% acetonitrile (A) and 90% acetonitrile in deionized water (B). Gradient elution was used (0–5 min for 100% B, 5–28.5 min for 0% B, 28.5–33 min for 100% B). The measurement was performed at a temperature of 30 °C and a flow rate of 0.5 mL/min. The injection volume was 3 µL and the analysis time was 33 min (minute post time). For controlling the reproducibility of chromatographic analyses, lutein (444 nm, ≥ 95%, Labicom, Olomouc, Czech Republic, CAS# 127-40-2) and β-carotene (450 nm, ≥ 99%, Sigma-Aldrich, Taufkirchen, Germany, CAS# 7235-40-7) were used as standards.

### 4.10. Flow Cytometric Analysis of the Protective Effect of Crude Extract

The following human cell lines were tested: HEK-293, KYSE-180, HeLa, HEP-G2, MCF-7, LNCaP, and KYSE-30. Cell line culture media were prepared using sterile solutions of fetal bovine serum FBS (10%), L-Glutamine (1%), and cell culture medium RPMI 1640 (89%). Cells were grown in RPMI 1640 supplemented with 10% of FBS (Merck, Prague, Czech Republic) at 37 °C under a 5% CO_2_ atmosphere. A day before CE addition, the cells were seeded in a 24-well plate at a density 3.10^5^ cells/mL. 24 h later, two different amounts of CE in 0.5% and 1% dimethyl sulfoxide (DMSO) were added to the cells and incubation continued for 24 h. The final concentration of CE in wells was expressed as total cellular carotenoids (0.1 and 0.2 ng/µL). Then, 25 μL of a 20 μM probe solution (5.8 μg of general oxidative stress indicator CM-H2DCFDA in 1% DMSO and phosphate-buffered saline, PBS) was added dropwise to the cells. After 30 min incubation, hydrogen peroxide was added in increasing concentrations; 0, 500, 1000, 1500, 2000, and 2500 µM and incubation proceeded for another 60 min. The cells were then trypsinized and the population of fluorescent cells were determined at A_485_ nm by a flow cytometer (FACS Aria III from BD Biosciences, San Jose, CA, USA).

### 4.11. Testing for Cytotoxicity of the Crude Extract

The human cell lines LNCaP and KYSE-30 were used to test for cytotoxicity. Cell line culture medium was prepared using sterile solutions of FBS (10%), L-Glutamine (1%), cell culture medium RPMI 1640 (89%). LNCaP were grown in RPMI 1640 supplemented with 10% FBS (Merck, Prague, Czech Republic) and KYSE-30 in RPMI 1640 with HAM’s F12 1:1 supplemented with 10% FBS and 1% L -glutamine at 37 °C in an atmosphere of 5% CO_2_. One day before CE addition, the cells were seeded into a 48-well plate at a density of 3.10^5^ cells/mL. 24 h later, a defined amount of CE and β-carotene (BC) in 0.5%, 0.75%, and 1% DMSO and pure DMSO (0.5–1%) were added to the cells, and incubation proceeded for 24 h. The final concentrations of β-carotene in the wells were 0.1, 0.15, and 0.2 ng/µL. The final concentrations of CE in wells were expressed as TC (0.1, 0.15, and 0.2 ng/µL). Then, a sterile stock solution of resazurin (15 mg/mL in PBS) was added dropwise to the cells. After 4 h the cell populations were measured at 590 nm with excitation at 560 nm and carried out on a multifunction reader (Infinite M200PRO, Tecan Life Sciences, Männedorf, Switzerland).

### 4.12. Statistical Analysis

Data are presented as means ± standard deviations. Experimental data were statistically evaluated using ANOVA and linear/quadratic models. All statements of significance were based on a probability of *p* < 0.05. Statistical analyses were performed using the Statistica 12.0.

## 5. Conclusions

Physiological tools such as cultivation conditions enable us to modify the composition of microbial biomass. This approach for enhancing the production of desired microbial compounds can be as effective as genetic engineering, while being easier to implement. Both light intensity and temperature were able to modify the crude extract from *S. tropica* biomass towards higher total cellular carotenoid and sioxanthin contents, as well as antioxidant activity. The crude extract thus obtained showed the largest protective effect against oxidative stress of human cell lines while simultaneously having little or no cytotoxicity. Nevertheless, other unidentified compounds in the crude extract need to be characterized and their biological activity revealed.

## Figures and Tables

**Figure 1 marinedrugs-19-00509-f001:**
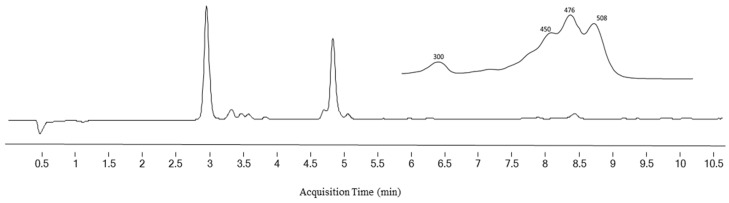
DAD chromatogram of a sample of crude *S. tropica* carotenoid extract measured at 450 nm on UHPLC/DAD/MSD Q-TOF system. Sioxanthin peak at RT 3.0 min. Insert figure is the UV–VIS spectrum of sioxanthin with three absorption maxima at 450, 476, and 508 nm.

**Figure 2 marinedrugs-19-00509-f002:**
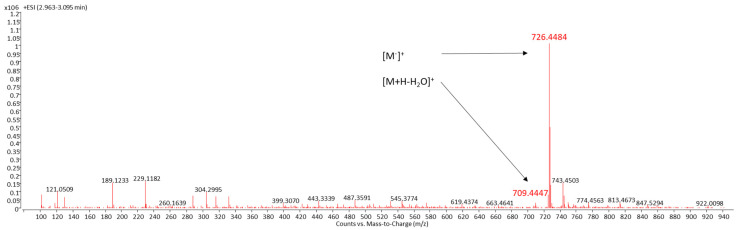
MS/MS spectrum of the sioxanthin peak at RT 3.0 min (Figure 1) with its molecular ions.

**Figure 3 marinedrugs-19-00509-f003:**
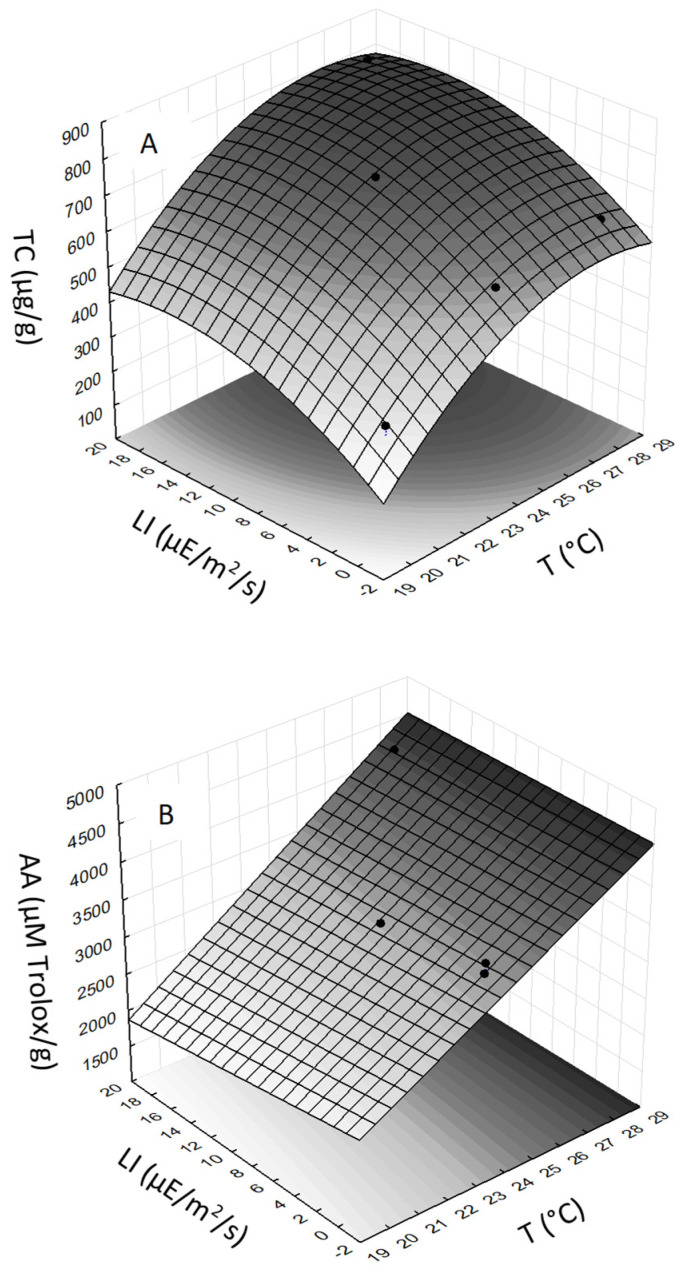
Response surfaces showing the effect of light intensity (LI) and temperature (T) on (**A**)—total cellular carotenoids (TC); (**B**)—antioxidant activity (AA); and (**C**)—sioxanthin (SX) peak area in crude extracts from S. *tropica*.

**Figure 4 marinedrugs-19-00509-f004:**
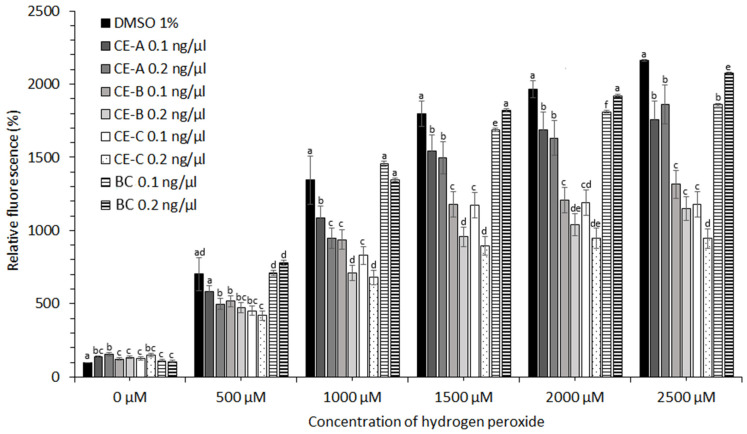
Relative fluorescence of oxidative stress indicator in LNCaP cell lines in response to the protective effect of carotenoid extracts (CE-A, CE-B, and CE-C) from *S. tropica* biomass and of β-carotene (BC) in the presence of different H_2_O_2_ concentrations. Both CE and β-carotene at 0.1 and 0.2 ng/µL were dissolved in 0.5 and 1% DMSO, respectively. Means with at least one letter the same are not significantly different (*p* > 0.05). Comparison of statistical significance is presented separately for each H_2_O_2_ concentration. Error bars indicate the standard deviations.

**Figure 5 marinedrugs-19-00509-f005:**
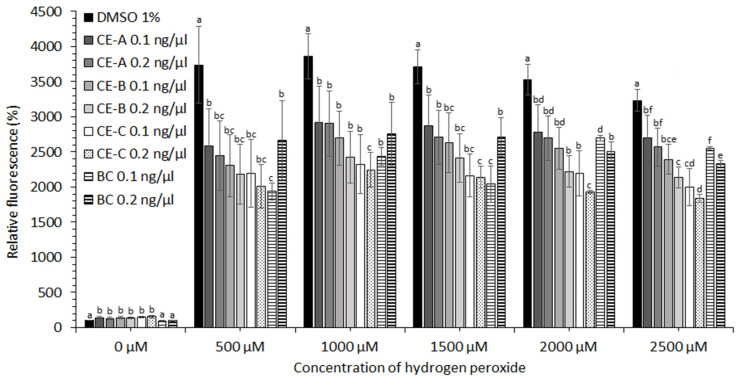
Relative fluorescence of oxidative stress indicator in KYSE-30 cell lines in response to the protective effect of carotenoid extracts (CE-A, CE-B, and CE-C) from *S. tropica* biomass and of β-carotene (BC) in the presence of different H_2_O_2_ concentrations. Both CE and β-carotene at 0.1 and 0.2 ng/µL were dissolved in 0.5% and 1% DMSO, respectively. Means with at least one letter the same are not significantly different (*p* > 0.05). Comparison of statistical significance is presented separately for each H_2_O_2_ concentration. Error bars indicate the standard deviations.

**Table 1 marinedrugs-19-00509-t001:** Effect of cultivation temperature and light intensity on the characteristics of the crude extracts from *S. tropica* biomass: total cellular carotenoid content, antioxidant activity and sioxanthin peak area. The results are average values from shake flask experiments carried out in triplicate.

Temperature (°C)	Light Intensity (µE/m^2^/s)	Total Cellular Carotenoids (µg/g)	Antioxidant Activity (µM Trolox/g)	Sioxanthin Peak Area
20 ^a^	0 ^a^	354 ± 2.5	2435.3 ± 15.6	49.6 ± 1.5
24	0	584 ± 13	3623.0 ± 39.9	124.8 ± 4.1
24 ^b^	9.30 ^b^	739 ± 35	3324.8 ± 13.2	204.5 ± 3.6
24	0	581 ± 5	3487.1 ± 30.9	136.0 ± 2.9
28	0	628 ± 30	4208.3 ± 12.0	208.4 ± 2.9
28 ^c^	18.60 ^c^	800 ± 8	4259.2 ± 25.7	502.5 ± 3.3

^a^ Biomass from this cultivation was used to prepare CE-A. ^b^ Biomass from this cultivation was used to prepare CE-B. ^c^ Biomass from this cultivation was used to prepare CE-C.

## Data Availability

The data presented in this study are available on request from the corresponding author.

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
