# Peer review of "Influence of Cultivation Conditions on the Sioxanthin Content and Antioxidative Protection Effect of a Crude Extract from the Vegetative Mycelium of Salinispora tropica"

_marinedrugs, 2021, doi:10.3390/md19090509_

Round 1

Reviewer 1 Report

The author addressed revision appropriately.

Author Response

No comments to be addressed.

Reviewer 2 Report

The manuscript by Jeskova et al., should be acceptable for publication in this journal. Overall, this study appears interesting for the natural product community, especially for the Actinobacteria group. I have only a few minor comments and suggestions for this paper. These are specifically addressed below.

Line 41: Indicate the superscript "T" for S. tropica strain.

Line 122 (Table 1): I think it would be nice to also indicate the total biomass yield in each condition so there will be a correlation on the growth of the strain.

Line 199: In Synechocystis sp., "sp." should not be italicized here and elsewhere (e.g. Line 217, Formosa sp., etc.) in the manuscript.

Line 248: The corresponding DSM number should also be indicated for strain CBN-440T.

Line 253: The cell lines should have also a corresponding ATCC accession number and should be indicated as well.

Line 260: The brand of bacteriological agar should be included.

Line 274: I was really wondering why the biomass was washed with distilled water. I thought S. tropica CBN-440T is a marine-derived actinobacterium and sea salt solution should have been used instead of water. Am not sure if the cells were lysed during this process of washing. In addition, the volume used for washing should be included.

Line 288: Should it be more appropriate and precise to call Total Cellular Carotenoids instead of just Total Carotenoids (TC)?

Line 311 (Section 4.9): This section tells about more of the HPLC condition and measurements. Although the mass, retention time and the standard were mentioned in the result section, I think it is more appropriate to place them here as the title says "Determination of Sioxanthin."

Line 313: Indicate the amount of the acetone/methanol ratio.

Line 369: The "Patents" heading is I guess inappropriate.

Authors should also check carefully the manuscript for some typographical mistakes (e.g. Line 140, two punctuations after the word concentrations).

Author Response

Line 41: Indicate the superscript "T" for S. tropica strain.

Answer: corrected

Line 122 (Table 1): I think it would be nice to also indicate the total biomass yield in each condition so there will be a correlation on the growth of the strain.

Answer: These values are unfortunately not available. The biomass concentration was not determined, because we needed all the biomass for further experiments. The scale of the cultivations was150 mL.

Our previous article deals with cultivation and biomass yields: Jezkova, Z.; Binda, E.; Potocar, T.; Marinelli, F.; Halecky, M.; Branyik, T. Laboratory scale cultivation of Salinispora tropica in shake flasks and mechanically stirred bioreactors. Biotechnol. Lett. 2021, doi: 10.1007/s10529-021-03121-1

Line 199: In Synechocystis sp., "sp." should not be italicized here and elsewhere (e.g. Line 217, Formosa sp., etc.) in the manuscript.

Answer: corrected

Line 248: The corresponding DSM number should also be indicated for strain CBN-440T.

Answer: corrected

Line 253: The cell lines should have also a corresponding ATCC accession number and should be indicated as well.

Answer: corrected

Line 260: The brand of bacteriological agar should be included.

Answer: corrected

Line 274: I was really wondering why the biomass was washed with distilled water. I thought S. tropica CBN-440T is a marine-derived actinobacterium and sea salt solution should have been used instead of water. Am not sure if the cells were lysed during this process of washing. In addition, the volume used for washing should be included.

Answer: The biomass was washed with distilled water to remove excess salts that affect the biomass weight after lyophilization. There was a short contact time with distilled water with no effect on cell integrity. The biomass in pellets was observed microscopically.

Line 288: Should it be more appropriate and precise to call Total Cellular Carotenoids instead of just Total Carotenoids (TC)?

Answer: The term total cellular carotenoids was corrected on lines 17, 26, 62, 123, table 1 (line 125), 135, 293, 294, 339 and 371. The abbreviation TC has been retained in the text.

Line 311 (Section 4.9): This section tells about more of the HPLC condition and measurements. Although the mass, retention time and the standard were mentioned in the result section, I think it is more appropriate to place them here as the title says "Determination of Sioxanthin."

Answer: Chapter 4.9 deals only with the technical aspect of the determination of sioxanthin, as it is a chapter in the methodology section. The mass and retention time are placed among results because it relates to the following section (2.2) on the effect of culture conditions on the composition of the crude extract, including the sioxanthin content.

Line 313: Indicate the amount of the acetone/methanol ratio.

Answer: corrected

Line 369: The "Patents" heading is I guess inappropriate.

Answer: deleted

Authors should also check carefully the manuscript for some typographical mistakes (e.g. Line 140, two punctuations after the word concentrations).

Answer: checked